# Faster Maximum Inner Product Search in High Dimensions

## Abstract

Maximum Inner Product Search (MIPS) is a ubiquitous task in machine learning applications such as recommendation systems. Given a query vector and $n$ atom vectors in $d$-dimensional space, the goal of MIPS is to find the atom that has the highest inner product with the query vector. Existing MIPS algorithms scale at least as $O(\sqrt{d})$, which becomes computationally prohibitive in high-dimensional settings that are prevalent in various real-world scenarios. In this work, we present BanditMIPS, a novel randomized MIPS algorithm whose complexity is independent of $d$. BanditMIPS estimates the inner product for each atom by adaptively subsampling coordinates for more promising atoms, a strategy motivated by multi-armed bandits. We provide theoretical guarantees that BanditMIPS returns the correct answer with high probability, while improving the complexity in $d$ from $O(\sqrt{d})$ to $O(1)$. We also perform experiments on four synthetic and real-world datasets and demonstrate that BanditMIPS outperforms prior state-of-the-art algorithms. For example, in the Movie Lens dataset ($n$=4,000, $d$=6,000), BanditMIPS is 20× faster than the next best algorithm while returning the same answer. BanditMIPS requires no preprocessing of the data and includes a hyperparameter that practitioners may use to trade off accuracy and runtime. We also propose a variant of our algorithm, named BanditMIPS-$\alpha$, which achieves further speedups by employing non-uniform sampling across coordinates, and demonstrate how known preprocessing techniques can be used to further accelerate BanditMIPS. Finally, we illustrate the potential of BanditMIPS as a versatile subroutine, enabling any machine learning algorithms that employ MIPS (e.g. Matching Pursuit, Hierarchical Navigable Small Worlds, and a classification layer in a large language model) to harness rich high-dimensional datasets without the need for dimensionality reduction.

## 1 Introduction

The Maximum Inner Product Search problem (MIPS) (Shrivastava and Li 2014a; Neyshabur and Srebro 2015; Yu et al. 2017) is a ubiquitous task that arises in many machine learning applications, such as matrix-factorization-based recommendation systems (Koren, Bell, and Volinsky 2009; Cremonesi, Koren, and Turrin 2010), multi-class prediction (Dean et al. 2013; Jain and Kapoor 2009), structural SVM (Joachims 2006; Joachims, Finley, and Yu 2009), and computer vision (Dean et al. 2013). Given a *query* vector $\mathbf{q} \in \mathbb{R}^d$

and $n$ *atom* vectors $\mathbf{v}_1, \ldots, \mathbf{v}_n \in \mathbb{R}^d$, MIPS aims to find the atom most similar to the query:

$$i^* = \underset{i \in \{1, \cdots, n\}}{\arg \max} \mathbf{v}_i^T \mathbf{q} \qquad (1)$$

For example, in recommendation systems, the query $\mathbf{q}$ may represent a user and the atoms $\mathbf{v}_i$'s represent items with which the user can interact; MIPS finds the best item for the user, as modeled by their concordance $\mathbf{v}_i^T \mathbf{q}$ (Amagata and Hara 2021; Aouali et al. 2022). In many applications, the number of atoms $n$ and the feature dimension $d$ can easily be in the millions, so it is critical to solve MIPS accurately and efficiently (Hirata et al. 2022).

The naïve approach evaluates all $nd$ elements and scales as $O(nd)$. Prior methodologies have aimed to reduce scaling with $n$ by reconstructing the underlying data structure which requires heavy preprocessing, especially on datasets with large $d$ (Morozov and Babenko 2018a; Liu et al. 2020). To avoid this overhead, more recent works such as (Lorenzen and Pham 2021) and (Liu, Wu, and Mozafari 2019) advocate for sampling-based approaches. However, the complexity remains at best $O(\sqrt{d})$, which is not ideal, considering the prevalence of high dimensional datasets in domains such as e-commerce, genomics, and finance. Current practices attempt to address this issue with dimensionality reduction techniques, but this induces information loss (particularly in higher dimensions) and tends to scale with $O(d)$(Li and Wan 2020).

To this end, we propose BanditMIPS, a state-of-the-art randomized algorithm that solves MIPS problems on the fly. We demonstrate BanditMIPS 's dimensionality-independent complexity and provide a tunable hyperparameter that governs the tradeoff between accuracy and speed, a need identified by previous works (Yu et al. 2017). We also provide theoretical guarantees that BanditMIPS recovers the exact solution to Equation (1) with high probability in $\tilde{O}(\frac{n}{\Delta^2})$[*] time, where $\Delta$ is an instance-specific factor that does not depend on $d$. We have also performed comprehensive experiments to evaluate our algorithm's performance in two synthetic and two real-world datasets. For example, in the Movie Lens dataset ($n = 4,000, d = 6,000$) (Harper and Konstan 2015), BanditMIPS is 20× faster than prior state-of-the-art while returning the same answer.

---

[*]The $\tilde{O}$ notation hides logarithmic factors.

At a high-level, instead of computing the inner product $\mathbf{v}_i^T \mathbf{q}$ for each atom $i$ using all $d$ coordinates, BanditMIPS estimates them by subsampling a subset of coordinates. Since more samples give higher estimation accuracy, BanditMIPS adaptively samples more coordinates for top atoms to discern the best atom. The specific adaptive sampling procedure is motivated by multi-armed bandits (MAB) (Even-Dar, Mannor, and Mansour 2006).

BanditMIPS is easily parallelizable and can be used with other optimization objectives that decompose coordinate-wise. Unlike previous works, it does not require preprocessing, dimensionality reduction, or normalization of the data, nor does it require the query or atoms to be nonnegative (Yu et al. 2017). This versatility allows BanditMIPS to work as a flexible subroutine for various machine learning tasks and also lends itself to specific extensions as shown below. In summary, our work offers the following contributions:

- **Novel Algorithm: BanditMIPS .** We introduce BanditMIPS , a new state-of-the-art algorithm for MIPS in high-dimensional settings. Achieving $O(1)$ sample complexity with respect to dimensionality eliminates the need for preprocessing and dimensionality reduction techniques, empowering rich high-dimensional data processing.

- **Three Algorithmic Extensions.** First, we propose BanditMIPS-$\alpha$, which provides additional runtime speedups by sampling coordinates intelligently (Section 3). Second, we extend BanditMIPS to find the $k$ atoms with the highest inner products with the query ($k$-MIPS) in our experiments (Section 5 and Appendix **??**). Third, we discuss how BanditMIPS can be used in conjunction with preprocessing techniques leading to complexity reductions in both the number $n$ and dimension $d$ of the dataset (Appendix 5).

- **Versatile Integration**. We accelerate machine learning applications such as Orthogonal Matching Pursuit and a classification layer of a large language model using BanditMIPS as a black-box subroutine (Appendix 10).

- **Empirical Superiority**. We demonstrate that BanditMIPS outperforms rivals, achieving up to $30\times$ efficiency over the next best algorithm due to reduced sample usage in real datasets.

## Related work

**MIPS applications:** MIPS arises naturally in many information retrieval contexts (Sivic and Zisserman 2003; Dong et al. 2012; Boytsov et al. 2016) and for augmenting large, auto-regressive language models (Borgeaud et al. 2022). MIPS is also a subroutine in the Matching Pursuit problem (MP) and its variants, such as Orthogonal Matching Pursuit (OMP) (Locatello et al. 2017). MP and other iterative MIPS algorithms have found many applications, e.g., to find a sparse solution of underdetermined systems of equations (Donoho et al. 2012) and accelerate conditional gradient methods (Song et al. 2022; Xu, Song, and Shrivastava 2021). MIPS also arises in the inference stages of many other applications, such as for deep-learning based multi-class or multi-label classifiers (Dean et al. 2013; Jain and Kapoor 2009) and has been used as a black-box subroutine to improve the learning

and inference in unnormalized log-linear models when computing the partition function is intractable (Mussmann and Ermon 2016).

**MIPS algorithms:** Many approaches focus on solving approximate versions of MIPS. Such work often assumes that the vector entries are nonnegative, performs non-adaptive sampling (Lu, Hu, and Zeng 2017; Ballard et al. 2015; Lorenzen and Pham 2021; Ding, Yu, and Hsieh 2019; Yu et al. 2017), or rely on product quantization (Dai et al. 2020; Wu et al. 2019; Guo et al. 2020, 2019; Matsui et al. 2018; Douze, Jégou, and Perronnin 2016; Ge et al. 2013; Babenko and Lempitsky 2012; Jégou et al. 2011; Jégou, Douze, and Schmid 2011). Many of these algorithms require significant preprocessing, are limited in their adaptivity to the underlying data distribution, provide no theoretical guarantees, or scale linearly in $d$—all drawbacks that have been identified as bottlenecks for MIPS in high dimensions (Ponomarenko et al. 2014).

A large family of MIPS algorithms are based on locality-sensitive hashing (LSH) (Indyk and Motwani 1998; Shrivastava and Li 2014a, 2015; Neyshabur and Srebro 2015; Huang et al. 2015; Song et al. 2021; Lu and Kudo 2021; Shrivastava and Li 2014b; Wu et al. 2022; Huang et al. 2018; Ma et al. 2021; Andoni et al. 2015; Yan et al. 2018). A shortcoming of these LSH-based approaches is that, in high dimensions, the maximum dot product is often small compared to the vector norms, which necessitates many hashes and significant storage space (often orders of magnitude more than the data itself). Many other MIPS approaches are based on proximity graphs, such as ip-NSW (Morozov and Babenko 2018a) and related work (Liu et al. 2020; Feng et al. 2023; Tan et al. 2019, 2021; Zhou et al. 2019; Chen et al. 2022; Zhang, Wang, and He 2022; Alexander et al. 2011; Malkov and Yashunin 2016; Malkov et al. 2014). These approaches use preprocessing to build an index data structure that allows for more efficient MIPS solutions at query time. However, these approaches also do not scale well to high dimensions as the index structure (an approximation to the true proximity graph) is subject to the curse of dimensionality (Liu et al. 2020).

Perhaps most similar to our work is BoundedME which solves the MIPS problem based on an adaptive sampling approach (Liu, Wu, and Mozafari 2019). However, this method scales as $O(n\sqrt{d})$ which is objectively inferior to Bandit-MIPS 's independence with dimension $d$. The worse scaling comes from predetermining the number of times each atom is sampled by $d$ and not adapting to the actual *values* of the sampled inner products as done in BanditMIPS; rather, BoundedME is only adaptive to the relative *ranking* of the inner products. Intuitively, this approach is wasteful because information contained in the sampled inner product's values is discarded. Additional related work is discussed in Appendix 7.

**Multi-armed bandits:** BanditMIPS is motivated by the best-arm identification problem in multi-armed bandits (Even-Dar, Mannor, and Mansour 2006; Karnin, Koren, and Somekh 2013; Audibert, Bubeck, and Munos 2010; Jamieson and Nowak 2014; Jamieson et al. 2014; Jamieson and Talwalkar 2016; Bubeck, Munos, and Stoltz 2011; Bardenet and Mail-

lard 2015; Boucheron, Lugosi, and Massart 2013; Even-Dar, Mannor, and Mansour 2002; Kalyanakrishnan et al. 2012). In a typical setting, we have $n$ arms each associated with an expected reward $\mu_i$. At each time step $t = 0, 1, \cdots$, we decide to pull an arm $A_t \in \{1, \cdots, n\}$, and receive a reward $X_t$ with $E[X_t] = \mu_{A_t}$. The goal is to identify the arm with the largest reward with high probability while using the fewest number of arm pulls. The use of MAB-based adaptive sampling to develop computationally efficient algorithms has seen many applications, such as random forests and $k$-medoid clustering (Tiwari et al. 2020; Bagaria et al. 2018; Bagaria, Kamath, and Tse 2018; Zhang, Zou, and Tse 2019a; Bagaria et al. 2021).

## 2 Preliminaries and Notation

We consider a query $\mathbf{q} \in \mathbb{R}^d$ and $n$ atoms $\mathbf{v}_1, \ldots, \mathbf{v}_n \in \mathbb{R}^d$. Let $[n]$ denote $\{1, \ldots, n\}$, $q_j$ the $j$th element of $\mathbf{q}$, and $v_{ij}$ the $j$th element of $\mathbf{v}_i$. For a given query $\mathbf{q} \in \mathbb{R}^d$, the MIPS problem is to find the solution to Equation (1): $i^* = \arg\max_{i \in [n]} \mathbf{v}_i^T \mathbf{q}$.

We let $\mu_i := \frac{\mathbf{v_i}^T \mathbf{q}}{d}$ denote the *normalized inner product* for atom $\mathbf{v}_i$. Since the inner products $\mathbf{v_i}^T \mathbf{q}$ tend to scale linearly with $d$ (e.g., if each coordinate of the atoms and query are drawn i.i.d.), each $\mu_i$ should not scale with $d$. Note that $\arg\max_{i \in [n]} \mathbf{v}_i^T \mathbf{q} = \arg\max_{i \in [n]} \mu_i$ so it is sufficient to find the atom with the highest $\mu_i$. Furthermore, for $i \neq i^*$ we define the gap of atom $i$ as $\Delta_i := \mu_{i^*} - \mu_i \geq 0$ and the minimum gap as $\Delta := \min_{i \neq i^*} \Delta_i$. We primarily focus on the computational complexity of MIPS with respect to $d$.

## 3 Algorithm

---
**Algorithm 1** BanditMIPS
---
**Input**: Atoms $\mathbf{v}_1, \ldots, \mathbf{v}_n \in \mathbb{R}^d$, query $\mathbf{q} \in \mathbb{R}^d$, error probability $\delta$, sub-Gaussian parameter $\sigma$ **Output**: $i^* = \arg\max_{i \in [n]} \mathbf{q}^T \mathbf{v_i}$

1: $\mathcal{S}_{\text{solution}} \leftarrow [n]$
2: $d_{\text{used}} \leftarrow 0$
3: For all $i \in \mathcal{S}_{\text{solution}}$, initialize $\hat{\mu}_i \leftarrow 0, C_{d_{\text{used}}} \leftarrow \infty$
4: **while** $d_{\text{used}} < d$ and $|\mathcal{S}_{\text{solution}}| > 1$ **do**
5:      Sample a new coordinate $J \sim \text{Unif}[d]$
6:      **for all** $i \in \mathcal{S}_{\text{solution}}$ **do**
7:          $\hat{\mu}_i \leftarrow \frac{d_{\text{used}} \hat{\mu}_i + v_{iJ} q_J}{d_{\text{used}} + 1}$
8:          $\left(1 - \frac{\delta}{2nd_{\text{used}}^2}\right)$-CI: $C_{d_{\text{used}}} \leftarrow \sigma\sqrt{\frac{2\log\left(4nd_{\text{used}}^2/\delta\right)}{d_{\text{used}}+1}}$
9:      $\mathcal{S}_{\text{solution}} \leftarrow \{i : \hat{\mu}_i + C_{d_{\text{used}}} \geq \max_{i'} \hat{\mu}_{i'} - C_{d_{\text{used}}}\}$
10:      $d_{\text{used}} \leftarrow d_{\text{used}} + 1$
11: If $|\mathcal{S}_{\text{solution}}| > 1$, update $\hat{\mu}_i$ to be the exact value $\mu_i = \mathbf{v}_i^T q$ for each atom in $\mathcal{S}_{\text{solution}}$ using all $d$ coordinates
12: **return** $i^* = \arg\max_{i \in \mathcal{S}_{\text{solution}}} \hat{\mu}_i$

---

The BanditMIPS algorithm is described in Algorithm 1 and is motivated by best-arm identification algorithms. As summarized in Table 1, we can view each atom $\mathbf{v}_i$ as an arm with the arm parameter $\mu_i := \frac{\mathbf{v}_i^T \mathbf{q}}{d}$. When pulling an

arm $i$, we randomly sample a coordinate $J \sim \text{Unif}[d]$ and evaluate the inner product at the coordinate as $X_i = q_J v_{iJ}$. Using this reformulation, the best atom can be estimated using techniques from best-arm algorithms.

BanditMIPS can be viewed as a combination of UCB and successive elimination (Lai and Robbins 1985; Even-Dar, Mannor, and Mansour 2006; Zhang, Zou, and Tse 2019b). Algorithm 1 uses the set $\mathcal{S}_{\text{solution}}$ to track all potential solutions to Equation (1); $\mathcal{S}_{\text{solution}}$ is initialized as the set of all atoms $[n]$. We will assume that, for a fixed atom $i$ and a randomly sampled coordinate, the random variable $X_i = q_J v_{iJ}$ is $\sigma$-sub-Gaussian for some known parameter $\sigma$. With this assumption, Algorithm 1 maintains a mean objective estimate $\hat{\mu}_i$ and confidence interval (CI) for each potential solution $i \in \mathcal{S}_{\text{solution}}$, where the CI depends on the error probability $\delta$ as well as the sub-Gaussian parameter $\sigma$. We discuss the sub-Gaussian parameter and possible relaxations of this assumption in Subsections 3 and 4.

**Additional speedup techniques**

**Non-uniform sampling reduces variance:** In the original version of BanditMIPS, we sample a coordinate $J$ for all atoms in $\mathcal{S}_{\text{solution}}$ uniformly from the set of all coordinates $[d]$. However, some coordinates may be more informative of the inner product than others. For example, larger entries of $\mathbf{v}_i$ may contribute more to the inner product with $\mathbf{q}$. As such, we sample each coordinate $j \in [d]$ with probability $w_j \propto q_j^{2\beta}$ and $\sum_j w_j = 1$, and estimate the arm parameter $\mu_i$ of atom $i$ as $X = \frac{1}{w_J} q_J v_{iJ}$. $X$ is an unbiased estimator of $\mu_i$ and the specific choice of coordinate sampling weights minimizes the combined variance of $X$ across all atoms; different values of $\beta$ corresponds to the minimizer under different assumptions. We provide theoretical justification of this weighting scheme in Section 4. We note that the effect of this non-uniform sampling will only accelerate the algorithm.

**Warm start increases speed:** One may wish to perform MIPS for a batch of $m$ queries instead of just a single query, solving $m$ separate MIPS problems. In this case, we can cache the atom values for all atoms across a random subset of coordinates, and provide a warm start to BanditMIPS by using these cached values to update arm parameter estimates $\hat{\mu}_i$, $C_i$, and $\mathcal{S}_{\text{solution}}$ for all $m$ MIPS problems. Such a procedure will eliminate the obviously less promising atoms and avoid repeated sampling for each of the $m$ MIPS problems and increases computational efficiency. We note that, since the $m$ MIPS problems are independent, the theoretical guarantees described in Section 4 still hold across all $m$ MIPS problems simultaneously.

**Sub-Gaussian assumption and construction of confidence intervals**

Crucial to the accuracy of Algorithm 1 is the construction of the $(1 - \delta)$-CI based on the $\sigma$-sub-Gaussianity of each $X_i = q_J v_{iJ}$. We note that the requirement for $\sigma$-sub-Gaussianity is rather general. In particular, when the coordinate-wise products between the atoms and query are bounded in $[a, b]$, then each $X_i$ is $\frac{b^2 - a^2}{4}$-sub-Gaussian. This is commonly the case, e.g., in recommendation systems where user ratings

Table 1: MIPS as a best-arm identification problem.

| Terminology | Best-arm identification | MIPS |
|---|---|---|
| Arms | $i = 1, \ldots, n$ | Atoms $\mathbf{v}_1, \ldots, \mathbf{v}_n$ |
| Arm parameter $\mu_i$ | Expected reward $\mathbb{E}[X_i]$ | Average coordinate-wise product $\frac{\mathbf{v}_i^T \mathbf{q}}{d}$ |
| Pulling arm $i$ | Sample a reward $X_i$ | Sample a coordinate $J$ with reward $q_J v_{iJ}$ |
| Goal | Identify best arm with probability at least $1 - \delta$ | Identify best atom with probability at least $1 - \delta$ |

(each element of the query and atoms) are integers between 0 and 5, and we use this implied value of $\sigma$ in our experiments in Section 5.

The $\frac{b^2 - a^2}{4}$-sub-Gaussianity assumption allows us to compute $1 - \delta$ CIs via Hoeffding's inequality, which states that for any random variable $S_n = Y_1 + Y_2 + \ldots Y_n$ where each $Y_i \in [a, b]$

$$P(|S_n - \mathbb{E}[S_n]| > \epsilon) \leq \exp\left(\frac{-2\epsilon^2}{n(b-a)^2}\right).$$

Setting $\delta$ equal to the right hand side and solving for $\epsilon$ gives the width of the confidence interval. $\sigma = \frac{b^2 - a^2}{4}$ acts as a variance proxy used in the creation of the confidence intervals by BanditMIPS; smaller variance proxies should result in tighter confidence intervals and lower sample complexities and runtimes.

In other settings where the sub-Gaussianity parameter may not be known *a priori*, it can be estimated from the data or the CIs can be constructed using the empirical Bernstein inequality instead (Maurer and Pontil 2009).

## 4  Theoretical Analysis

**Analysis of the Algorithm:** For Theorem 1, we assume that, for a fixed atom $\mathbf{v}_i$ and $d_{\text{used}}$ randomly sampled coordinates, the $(1 - \delta')$ confidence interval scales as $C_{d_{\text{used}}}(\delta') = O\left(\sqrt{\frac{\log 1/\delta'}{d_{\text{used}}}}\right)$ (note that we use $d_{\text{used}}$ and $\delta'$ here because we have already used $d$ and $\delta$). We note that the sub-Gaussian CIs satisfy this property, as described in Section 3.

**Theorem 1.** *Assume $\exists \, c_0 > 0$ s.t. $\forall \, \delta' > 0$, $d_{used} > 0$, $C_{d_{used}}(\delta') < c_0 \sqrt{\frac{\log 1/\delta'}{d_{used}}}$. With probability at least $1 - \delta$, BanditMIPS returns the correct solution to Equation (1) and uses a total of $M$ computations, where*

$$M \leq \sum_{i \in [n], i \neq i^*} \min\left[\frac{16 c_0^2}{\Delta_i^2} \log\left(\frac{n}{\delta \Delta_i}\right) + 1, 2d\right]. \quad (2)$$

Theorem 1 is proven in the appendices. We note that $c_0$ is the sub-Gaussianity parameter described in Section 3 and is a constant. Intuitively, Theorem 1 states that with high probability, BanditMIPS returns the atom with the highest inner product with $\mathbf{q}$. The instance-wise bound Equation (2) suggests the computational cost of a given atom $\mathbf{v}_i$, i.e., $\min\left[\frac{16 c_0^2}{\Delta_i^2} \log\left(\frac{n}{\delta \Delta_i}\right) + 1, 2d\right]$, depends on $\Delta_i$, which measures how close its optimization parameter $\mu_i$ is to $\mu_{i^*}$. Most

reasonably different atoms $i \neq i^*$ will have a large $\Delta_i$ and incur an $O\left(\frac{1}{\Delta^2} \log \frac{n}{\delta \Delta_i}\right)$ computation that is independent of $d$ when $d$ is sufficiently large.

Important to Theorem 1 is the assumption that we can construct $(1 - \delta')$ CIs $C_i(d_{\text{used}}, \delta')$ that scale as $O(\sqrt{\frac{\log 1/\delta'}{d_{\text{used}}}})$. As discussed in Section 3, this is under general assumptions, for example when the estimator $X_i = q_J v_{iJ}$ for each arm parameter $\mu_i$ has finite first and second moments (Catoni 2012) or is bounded.

Since each coordinate-wise multiplication only incurs $O(1)$ computational overhead to update running means and confidence intervals, sample complexity bounds translate directly to wall-clock times bounds up to constant factors. For this reason, our approach of focuses on sample complexity bounds, in line with prior work (Tiwari et al. 2020; Bagaria, Kamath, and Tse 2018).

**Discussion of the hyperparameter $\delta$:** The hyperparameter $\delta$ allows users to trade off accuracy and runtime when calling Algorithm 1. A smaller value of $\delta$ corresponds to a lower error probability, but will lead to longer runtimes because the confidence intervals constructed by Algorithm 1 will be wider and atoms will be filtered more slowly. Theorem 1 provides an analysis of the effect of $\delta$ and in Section 5, we discuss appropriate ways to tune it. We note that setting $\delta = 0$ reduces Algorithm 1 to the naïve algorithm for MIPS. In particular, Algorithm 1 is never worse in big-$O$ sample complexity than the naïve algorithm.

**Discussion of the importance of $\Delta$:** In general, Bandit-MIPS takes only $O\left(\frac{1}{\Delta^2} \log \frac{n}{\delta \Delta}\right)$ computations per atom if there is reasonable heterogeneity among them. As proven in Appendix 2 in (Bagaria et al. 2018), this is the case under a wide range of distributional assumptions on the $\mu_i$'s, e.g., when the $\mu_i$'s follow a sub-Gaussian distribution across the atoms. These assumptions ensure that BanditMIPS has an overall complexity of $O\left(\frac{n}{\Delta^2} \log \frac{n}{\delta \Delta}\right)$ that is independent of $d$ when $d$ is sufficiently large and $\Delta$ does not depend on $d$.

At first glance, the assumption that each $\Delta_i$ (and therefore $\Delta$) does not depend on $d$ may seem restrictive. However, such an assumption actually applies under a reasonable number of data-generating models. For example, if the atoms' coordinates are drawn from a latent variable model, i.e., the $\mu_i$'s are fixed in advance and the atoms' coordinates correspond to instantiations of a random variable with mean $\mu_i$, then $\Delta_i$ will be independent of $d$. As a concrete example, two users' $0/1$ ratings of movies may agree on 60% of movies and their atoms' coordinates correspond to observations of a

Bernoulli random variable with parameter 0.6. Other recent works provide further discussion on the conversion between an instance-wise bound like Equation (2) and an instance-independent bound that is independent of $d$ (Bagaria et al. 2018; Baharav and Tse 2019; Tiwari et al. 2020; Bagaria et al. 2021; Baharav et al. 2022).

However, we note that in the worst case BanditMIPS may take $O(d)$ computations per atom when most atoms are equally good, for example in datasets where the atoms are symmetrically distributed around **q**. For example, if each atom's coordinates are drawn i.i.d. from the *same* distribution, then the gaps $\Delta_i$ will scale inversely with $d$; to address this concern, we demonstrate how our algorithm maintains $O(1)$ scaling with respect to $d$ in practice in Appendix 11.

**Optimal weights for non-uniform sampling:** Let $J \sim P_{\mathbf{w}}$ be a random variable following the categorical distribution $P_{\mathbf{w}}$, where $\mathbb{P}(J = j) = w_j \geq 0$ and $\sum_{j \in [d]} w_j = 1$. The arm parameter $\mu_i$ of an atom $i$ can be estimated by the unbiased estimator $X_{iJ} = \frac{1}{d w_J} v_{iJ} q_J$. (Note that $d$ is fixed and known in advance). To see that $X_{iJ}$ is unbiased, we observe that $\mathbb{E}_{J \sim P_{\mathbf{w}}}[X_{iJ}] = \sum_{j \in [d]} w_j \frac{1}{d w_j} v_{ij} q_j = \sum_{j \in [d]} \frac{v_{ij} q_j}{d} = \mu_i$.

We are interested in finding the best weights $\mathbf{w}^*$, i.e., those that minimize the combined variance

$$\underset{w_1, \ldots, w_d \geq 0}{\arg \min} \sum_{i \in [n]} \mathrm{Var}_{J \sim P_{\mathbf{w}}}[X_{iJ}], \quad s.t. \sum_{j \in [d]} w_j = 1. \quad (3)$$

**Theorem 2.** *The solution to Problem (3) is*

$$w_j^* = \frac{\sqrt{q_j^2 \sum_{i \in [n]} v_{ij}^2}}{\sum_{j \in [d]} \sqrt{q_j^2 \sum_{i \in [n]} v_{ij}^2}}, \quad for \ j = 1, \ldots, d. \quad (4)$$

The proof of Theorem 2 is provided in Appendix 8.

**Remark:** In practice, computing the atom variance $\sum_{i \in [n]} v_{ij}^2$ requires $O(nd)$ operations and can be computationally prohibitive. However, we may approximate $\sum_{i \in [n]} v_{ij}^2$ based on domain-specific assumptions. Specifically, if we assume that for each coordinate $j$, $q_j$ has a similar magnitude as $v_{ij}$'s, we can approximate $\frac{1}{n} \sum_{i \in [n]} v_{ij}^2 \approx q_j^2$ and set $w_j^* = \frac{q_j^2}{\sum_{j \in [d]} q_j^2}$. In the non-uniform sampling versions of BanditMIPS, we use an additional hyperparameter $\beta$ and let $w_j^* \propto q_j^{2\beta}$. $\beta$ can be thought of as a temperature parameter which governs how uniformly (or not) we sample the coordinates based on the query vector's values. We note that $\beta = 1$ corresponds Equation (4).

The version we call BanditMIPS-$\alpha$ corresponds to taking the limit $\beta \to \infty$. In this case, we sort the query vector explicitly and sample coordinates in order of the sorted query vector; the sub-Gaussianity parameter used in BanditMIPS-$\alpha$ is then the same as that in the original problem with uniform sampling. While the sort incurs $O(d \log d)$ cost, we find this still improves the overall sample complexity of the algorithm relative to the closest baseline when $O(d \log d + n)$ is better than $O(n\sqrt{d})$, as is often the case in practice.

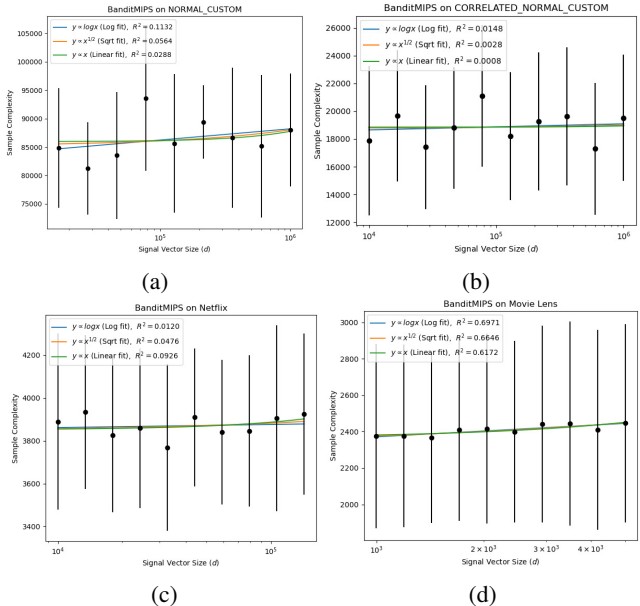

(a)          (b)

(c)          (d)

Figure 1: Sample complexity of BanditMIPS for different values of $d$ on all four datasets. 95% CIs are provided around the mean and are computed from 10 random trials. The sample complexity of BanditMIPS does not scale with $d$. Note that the values of $R^2$, the coefficient of determination, are similar for linear, logarithmic, and square root fits, which suggests the scaling is actually constant.

## 5 Experiments

We empirically evaluate the performance of BanditMIPS and the non-uniform sampling version BanditMIPS-$\alpha$ on four synthetic and real-world datasets, comparing them to 8 state-of-art MIPS algorithms. We considered the two synthetic datasets, NORMAL_CUSTOM and CORRELATED_NORMAL_CUSTOM, to assess the performance across a wide parameter range. We further considered the two real-world datasets, the Netflix Prize dataset ($n = 6,000$, $d = 400,000$) (Bennett, Lanning, and Netflix 2007) and the Movie Lens dataset ($n = 4,000$, $d = 6,000$) (Harper and Konstan 2015), to provide additional evaluations. We compared our algorithms to 8 baseline MIPS algorithms: LSH-MIPS (Shrivastava and Li 2014a), H2-ALSH-MIPS (Huang et al. 2018), NEQ-MIPS (Dai et al. 2020), PCA-MIPS (Bachrach et al. 2014), BoundedME (Liu, Wu, and Mozafari 2019), Greedy-MIPS (Yu et al. 2017), HNSW-MIPS (Malkov and Yashunin 2016; Morozov and Babenko 2018b). and NAPG-MIPS (Tan et al. 2021). Throughout the experiments, we focus on the sample complexity, defined as the number of coordinate-wise multiplications performed. Appendix 9 provides additional details on our experimental settings.

**Scaling with dimension $d$:** We first assess the scaling with $d$ for BanditMIPS on the four datasets. We subsampled features from the full datasets, evaluating $d$ up to $1,000,000$ on simulated data and up to $400,000$ on real-world data. Results are reported in Figure 1. In all trials, BanditMIPS returns the

correct answer to MIPS. We determined that BanditMIPS did not scale with $d$ in all experiments, validating our theoretical results on the sample complexity.

**Comparison of sample complexity:** We next compare the sample complexity of BanditMIPS and BanditMIPS-$\alpha$ to 8 state-of-art MIPS algorithms on the four datasets across different values of $d$. We only used a subset of up to 20K features because some of the baseline algorithms were prohibitively slow for larger values of $d$. Results are reported in Figure 2. We omit GREEDY-MIPS from Figure 2 because its sample complexity was significantly worse than all algorithms, and omit HNSW-MIPS as its performance was strictly worse than NAPG-MIPS (a related baseline). In measuring sample complexity, we measure *query-time* sample complexity, meaning we neglect the cost of preprocessing for the baseline algorithms which is favorable to the baselines. Nonetheless, our two algorithms substantially outperformed other algorithms on all four datasets, demonstrating their superiority in sample efficiency. For example, on the Movie Lens dataset, BanditMIPS and BanditMIPS-$\alpha$ are $20\times$ and $27\times$ faster than the closest baseline (NEQ-MIPS). In addition, the non-uniform sampling version BanditMIPS-$\alpha$ outperformed the default version BanditMIPS in 3 out 4 datasets, suggesting the weighted sampling technique further improves sample efficiency. BanditMIPS-$\alpha$ demonstrated slightly worse performance than BanditMIPS on the Netflix dataset, possibly because the highest-value coordinates for the randomly sampled query vectors had low dot products with the atoms.

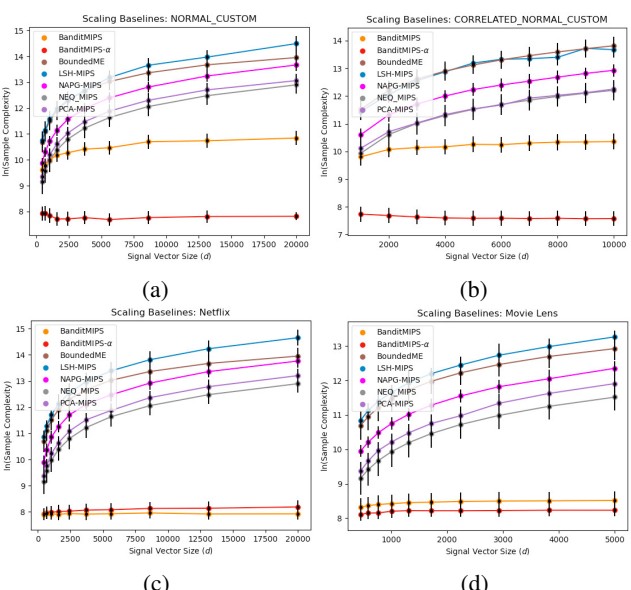

(a)      (b)

(c)      (d)

Figure 2: Comparison of sample complexity between BanditMIPS, BanditMIPS-$\alpha$, and other baseline algorithms for different values of $d$ across all four datasets. The $y$-axis is on a logarithmic scale. 95% CIs are provided around the mean and are computed from 10 random trials. BanditMIPS and BanditMIPS-$\alpha$ outperformed other baselines. For example, BanditMIPS achieves sample efficiency that surpasses the next best algorithm by up to $\times 30$ in the Movie Lens dataset.

| Algorithms | Speedup |
|---|---|
| Naïve algorithm | 1.00x |
| BoundedMe | 0.41x |
| BanditMIPS | 14.19x |
| BanditMIPS-$\alpha$ | 9.93x |

Table 2: Wall Clock Time Comparison on MNIST dataset. Experimental settings are as follows: $\epsilon$=0.1, $\delta$=0.1, number of atoms=1000, and signal vector size=6000.

| Algorithms | Speedup |
|---|---|
| Naïve algorithm | 1.00x |
| BoundedMe | 0.36x |
| BanditMIPS | 53.02x |
| BanditMIPS-$\alpha$ | 25.97x |

Table 3: Wall Clock Time Comparison on Movie Lens dataset. Experimental settings are as follows: $\epsilon$=0.1, $\delta$=0.1, number of atoms=1000, and signal vector size=100000.

| Algorithms | Speedup |
|---|---|
| Naïve algorithm | 1.00x |
| BoundedMe | 1.02x |
| BanditMIPS | 4.00x |
| BanditMIPS-$\alpha$ | 6.02x |

Table 4: Wall Clock Time Comparison on OPT-6.7B head dataset. OPT-6.7B head dataset refers to the MIPS task using the LM Head of the OPT-6.7B model as atoms and the final hidden vectors from randomly generated sentences as queries. Experimental settings are as follows: $\epsilon$=1.0, $\delta$=0.9, number of atoms=10000, and signal vector size=4096.

**Wallclock speedup and scaling:** We note that the sample complexity of BanditMIPS may not reliably predict its speedup for the MIPS problem, considering the highly optimized vector-vector dot product techniques. Thus, we offer wall-clock time comparisons as a complementary analysis, demonstrating that even with modest optimizations in Python, BanditMIPS significantly outperforms the next best algorithm (BoundedME). As shown in Table 2 and Table 3, BanditMIPS surpasses BoundedME by a large margin for the MNIST and Movie Lens dataset. We also employed BanditMIPS on the classification layer of OPT-6.7B where MIPS serves to efficiently determine the next token to generate. Table 4 demonstrates this impressive speedup achieved by BanditMIPS for dimension size 4096. All sampling-based algorithms successfully return an $\epsilon$-suboptimal atom with $1 - \delta$ probability for the given $\epsilon$ and $\delta$. Additionally, figure 3 demonstrates that BanditMIPS exhibits $O(1)$ scaling with respect to dimensionality ($d$) in terms of wall-clock time on the Netflix and Movie Lens dataset.

**Trade-off between speed and accuracy:** We evaluate the trade-off between speed and accuracy by varying the error probability $\delta$ in our algorithm and the corresponding hyper-parameters in the baseline algorithms (see Appendix 9 for more details). As in (Liu, Wu, and Mozafari 2019), we define the speedup of an algorithm to be:

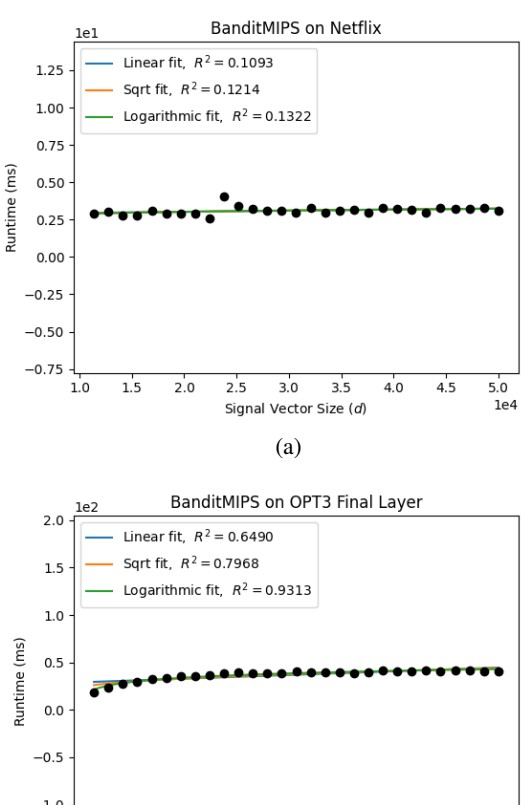

(a)

(b)

Figure 3: Wallclock time scaling of BanditMIPS for the Net-flix dataset and OPT3-6.7b LM Head (with OPT3-6.7b model as atoms and final hidden vectors from randomly generated sentences as queries). The runtime of BanditMIPS is constant in $d$, as is expected. Means were calculated from 10 random seeds. For the Netflix dataset we have $\epsilon = 0.1$, $\delta = 0.1$, and 1000 atoms. For the LM Head, settings are $\epsilon = 1.0$, $\delta = 0.9$, and 1000 atoms.

speedup $= \frac{\text{sample complexity of naïve algorithm}}{\text{sample complexity of compared algorithm}}$. The accuracy is defined as the proportion of times each algorithm returns the true MIPS solution. The tradeoff results for Normal Custom, Correlated Normal Custom, Netflix, and Movie Lens datasets are reported in Figure 4. Our algorithms achieved the best tradeoff on all four datasets, again demonstrating the superiority of our algorithms in efficiently and accurately solving the MIPS problem. Note that this figure also includes the $k$-MIPS setting where the goal is to find the top $k$ atoms. In our particular case, $k = 5$. Even in this setting, our algorithms obtained a similar improvement over other baselines.

**Real-world high-dimensional datasets:** We also verify the $O(1)$ scaling with $d$ on two real-world, high-dimensional datasets: the Sift-1M (Jégou, Douze, and Schmid 2011) and CryptoPairs datasets (Carsten 2022). The Sift-1M

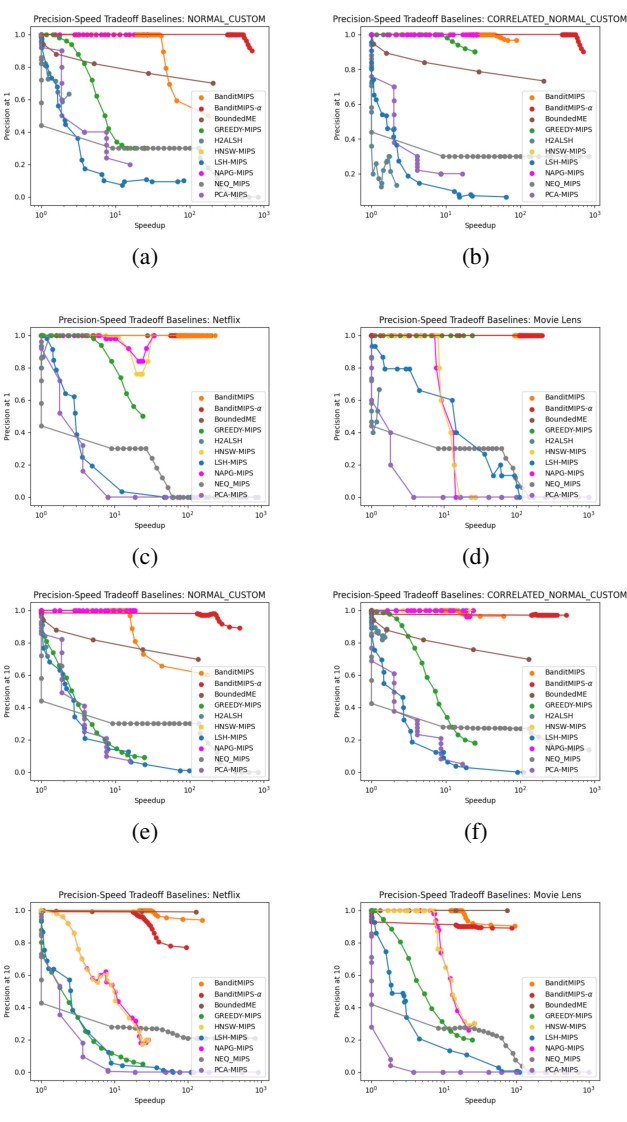

(a)      (b)

(c)      (d)

(e)      (f)

(g)      (h)

Figure 4: Trade-off between accuracy and speed for various algorithms across all four datasets. The $x$-axis represents the speedup relative to the naive $O(nd)$ algorithm and the $y$-axis shows the proportion of times an algorithm returned correct answer; higher is better. Each dot represents the mean across 10 random trials and the CIs are omitted for clarity. Our algorithms consistently achieve better accuracies at higher speedup values than the baselines. (a) through (d) is precision at $k = 1$ (i.e. the best arm) whereas (e) through (h) is precision at $k = 5$.

dataset consists of scale-invariant feature transform (Lowe 1999) features of 128 different images; each image is an atom with 1 million dimensions. The CryptoPairs dataset consists of the historical trading data of more than 400 trading pairs at 1 minute resolution reaching back until the year 2013. On these datasets, BanditMIPS appears to scale as $O(1)$ with $d$ even to a million dimensions (Figure 6). This suggests

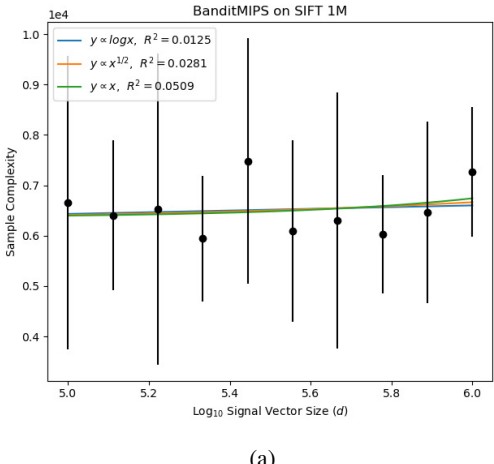

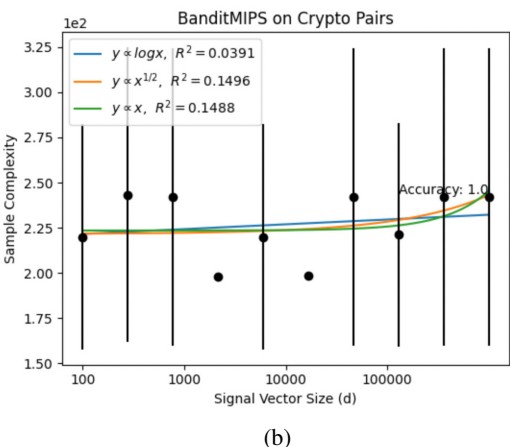

Figure 5: Sample complexity of BanditMIPS versus $d$ for the `Sift-1M` and `CryptoPairs` datasets. BanditMIPS scales as $O(1)$ with respect to $d$ for both datasets. Means and uncertainties were obtained by averaging over 5 random seeds. BanditMIPS returns the correct solution to MIPS in each trial.

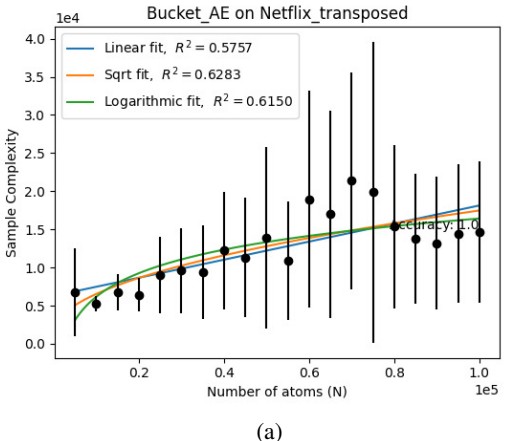

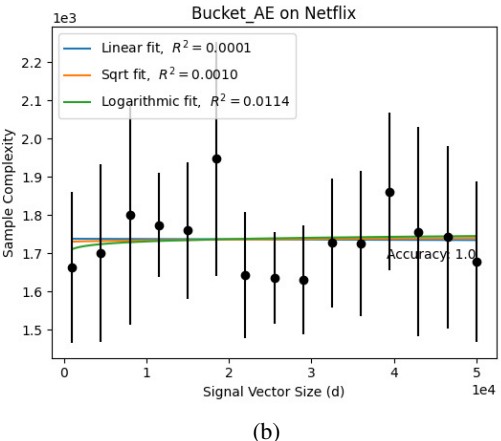

Figure 6: Sample complexity of `Bucket_AE` for both $n$ and $d$ on the `Netflix` dataset averaged over 5 random seeds. This demonstrates that BanditMIPS's constant scaling in dimension $d$ is independent of optimizations deployed in the $n$-dimension, opening the door for many extensions of BanditMIPS with existing techniques.

that the necessary assumptions outlined in Sections 3 and 4 are satisfied on these real-world, high-dimensional datasets. Note that the high dimensionality of these datasets makes them prohibitively expensive to run scaling experiments as in Section 5 or the tradeoff experiments as in Section 5 for baseline algorithms.

**Preprocessing with BanditMIPS:** BanditMIPS obviates any preprocessing in the dimension $d$ as shown in our experiments above. An added benefit is that our algorithm also works *in conjunction* with preprocessing methods that reduce the scaling with respect to the number of atoms $n$. To show this compatibility, we implemented `Bucket_AE` which combines BanditMIPS with a normalized binning technique. More precisely, we estimate the norm of each atom with a constant number of samples to which we sort them into bins of $b$ atoms in decreasing order ($b$ is a hyperparameter). When running BanditMIPS, we make comparisons between only the best atoms in each bin and eliminate an entire bin if the maximum potential of that bin's best atom is less than the current largest sampled inner product across all bins. Intuitively, this allows us to filter atoms with small estimated norm more quickly. Indeed, figure 6b demonstrates that `Bucket_AE` reduces the scaling with $n$ while maintaining $O(1)$ scaling with $d$ on the real-world Netflix dataset. Furthermore, we observed that `Bucket_AE` returns the correct solution to BanditMIPS for all trials.

**Robustness to $\epsilon$-corruption:** We end the experiments section with a discussion on BanditMIPS 's robustness to $\epsilon$-corruption. Most works in the MIPS literature do not analyze the case that the data is corrupted by noise and provide no guarantees about robustness in this setting. However, BanditMIPS is actually robust to $\epsilon$-corruption. More formally,

assume that each atom's coordinates are corrupted with noise drawn from a Gaussian with mean 0 and standard deviation $\epsilon$. By the Central Limit Theorem, this noise will effectively vanish when averaged across many coordinates. For this reason, in our implementation, we set the batch size (which is the minimum number of coordinates sampled for any atom) to 100 or greater. As long as $\frac{\epsilon}{\text{batch size}\sigma} << 1$, i.e., the noise level $\epsilon$ is reasonable, BanditMIPS will still return the correct solution to the MIPS problem. We verify this on the Netflix dataset corrupted under this noise model, where $\epsilon$ is set to 20% of the maximum possible coordinate value of the dataset. We observe that BanditMIPS still scales as $O(1)$ with respect to $d$ as shown in Figure 7 and returns correct solutions even at fairly high speedup values.

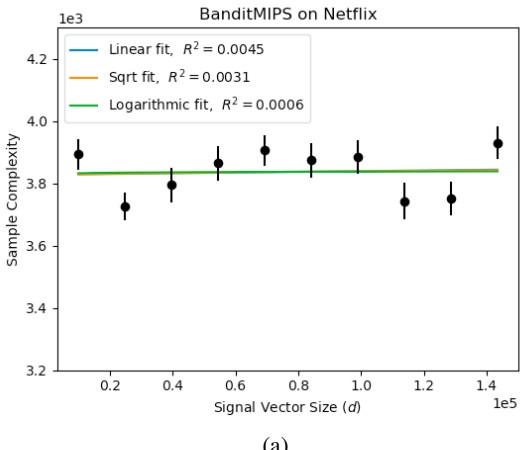

(a)

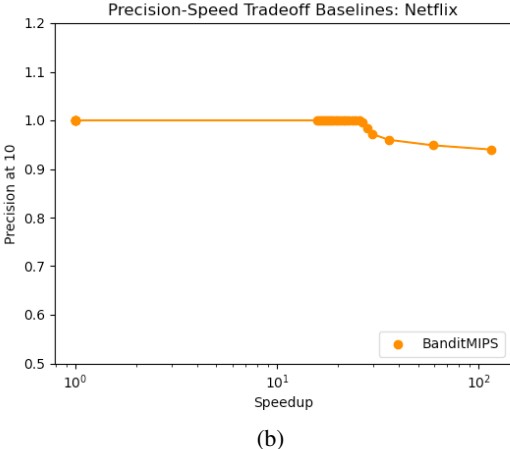

(b)

Figure 7: (a): Sample complexity of BanditMIPS versus dimension $d$ for the corrupted Netflix dataset. Even in the uncorrupted settings, BanditMIPS maintains its O(1) scaling with respect to dimensionality ($d$). (b): Trade-off between top-10 accuracy and speed. The $x$-axis represents the speedup relative to the naive $O(nd)$ algorithm and the $y$-axis shows the proportion of times an algorithm returned correct answer; higher is better. Our algorithm returns an accurate solution even at high speedup values.

# 6   Conclusions and Limitations

**Conclusions:** In this work, we presented BanditMIPS and BanditMIPS-$\alpha$, novel algorithms for the MIPS problem. In contrast with prior work, BanditMIPS requires no preprocessing of the data or that the data be nonnegative, and provides hyperparameters to trade off accuracy and runtime. BanditMIPS scales better to high-dimensional datasets under reasonable assumptions and outperformed the prior state-of-the-art significantly. BanditMIPS scales better to high-dimensional datasets under reasonable assumptions and outperformed the prior state-of-the-art significantly. Finally, we combined the approaches of BanditMIPS and BanditMIPS-$\alpha$ with other preprocessing techniques to reduce their scaling with $n$.

**Limitations:** Though the assumptions for BanditMIPS and BanditMIPS-$\alpha$ are often satisfied in practice, requiring them may be a limitation of our approach. In particular, when many of the arm gaps are small, BanditMIPS will compute the inner products for the relevant atoms naïvely.

**Future directions:** Future work may focus on relaxing these assumptions. Also, BanditMIPS has the potential to empower diverse machine learning algorithms with rich high-dimensional datasets, eliminating the necessity for dimensionality reduction. Subsequent exploration could delve into accelerating algorithms like Hierarchical Navigable Small World and a classification layer in a large language model.