## 7 Additional Related Work

In this appendix, we briefly describe other approaches attempt to reduce MIPS to a nearest neighbor search problem (NN). We note that the NN literature is extremely vast and has inspired the use of techniques based on permutation search (Naidan, Boytsov, and Nyberg 2015), inverted files (Amato and Savino 2008), vantage-point trees (Boytsov and Naidan 2013b), and more. The proliferation of NN algorithms has inspired several associated software packages (Bernhardsson 2018; Johnson, Douze, and Jégou 2019; Boytsov and Naidan 2013a) and tools for practical hyperparameter selection (Sun, Guo, and Kumar 2023). However, MIPS is fundamentally different from and harder than NN because the inner product is not a proper metric function (Morozov and Babenko 2018b). Nonetheless, NN techniques have inspired many direct approaches to MIPS, including those that rely on $k$-dimensional or random projection trees (Dasgupta 2008), concomitants of extreme order statistics (Pham 2020a, 2021, 2020b), ordering permutations (Chávez, Figueroa, and Navarro 2008), principle component analysis (PCA) (Bachrach et al. 2014), or hardware acceleration (Xiang et al. 2021; Abuzaid et al. 2019). All of these approaches require significant preprocessing that scales linearly in $d$, e.g., for computing the norms of the query or atom vectors, whereas BanditMIPS does not.

## 8 Proofs of Theorems

In this appendix, we present the proofs of Theorems 1 and 2.

**Proof of Theorem 1:**

*Proof.* Following the multi-armed bandit literature, we refer to each index $i$ as an arm and refer to its optimization object $\mu_i$ as the arm parameter. We sometimes abuse the terminology and refer to the atom $\mathbf{v}_i$ as the arm, with the meaning clear from context. Pulling an arm corresponds to uniformly sampling a coordinate $J$ and evaluating $v_{iJ}q_J$ and incurs an $O(1)$ computation. This allows us to focus on the number of arm pulls, which translates directly to coordinate-wise sample complexity.

First, we prove that with probability at least $1-\delta$, all confidence intervals computed throughout the algorithm are valid in that they contain the true parameter $\mu_i$'s. For a fixed atom $\mathbf{v}_i$ and a given iteration of the algorithm, the $\left(1 - \frac{\delta}{2nd_{\text{used}}^2}\right)$ confidence interval satisfies

$$\Pr\left(|\mu_i - \hat{\mu}_i| > C_{d_{\text{used}}}\right) \leq 2e^{-C_{d_{\text{used}}}^2 d_{\text{used}}/2\sigma^2} \leq \frac{\delta}{2nd_{\text{used}}^2}$$

by Hoeffding's inequality and the choice of $C_{d_{\text{used}}} = \sigma\sqrt{\frac{2\log(4nd_{\text{used}}^2/\delta)}{d_{\text{used}}+1}}$. For a fixed arm $i$, for any value of $d_{\text{used}}$ we

have that the confidence interval is correct with probability at least $1 - \frac{\delta}{n}$, where we used the fact that $1 + \frac{1}{2^2} + \frac{1}{3^2} + \ldots = \frac{\pi^2}{6} < 2$. By another union bound over all $n$ arm indices, all confidence intervals constructed by the algorithm are correct with probability at least $1 - \delta$.

Next, we prove the correctness of BanditMIPS. Let $i^* = \arg\max_{i\in[n]}\mu_i$ be the desired output of the algorithm. First, observe that the main `while` loop in the algorithm can only run $d$ times, so the algorithm must terminate. Furthermore, if all confidence intervals throughout the algorithm are valid, which is the case with probability at least $1 - \delta$, $i^*$ cannot be removed from the set of candidate arms. Hence, $\mathbf{v}_{i^*}$ (or some $\mathbf{v}_i$ with $\mu_i = \mu_{i^*}$) must be returned upon termination with probability at least $1 - \delta$. This proves the correctness of Algorithm 1.

Finally, we examine the complexity of BanditMIPS. Let $d_{\text{used}}$ be the total number of arm pulls computed for each of the arms remaining in the set of candidate arms at a given iteration in the algorithm. Note that for any suboptimal arm $i \neq i^*$ that has not left the set of candidate arms $\mathcal{S}_{\text{solution}}$, we must have $C_{d_{\text{used}}} \leq c_0\sqrt{\frac{\log(1/\delta)}{d_{\text{used}}}}$ by assumption (and this holds for our specific choice of $C_{d_{\text{used}}}$ in Algorithm 1). With $\Delta_i = \mu_{i^*} - \mu_i$, if $d_{\text{used}} > \frac{16c_0^2}{\Delta_i^2}\log\frac{n}{\delta\Delta_i}$, then

$$4C_{d_{\text{used}}} \leq 4c_0\sqrt{\frac{\log\frac{n}{\delta\Delta_i}}{d_{\text{used}}}} < \Delta_i$$

Furthermore,

$$\begin{aligned}
\hat{\mu}_{i^*} - C_{d_{\text{used}}} &\geq \mu_{i^*} - 2C_{d_{\text{used}}} \\
&= \mu_i + \Delta_i - 2C_{d_{\text{used}}} \\
&> \mu_i + 2C_{d_{\text{used}}} \\
&> \hat{\mu}_i + C_{d_{\text{used}}}
\end{aligned}$$

which means that $i$ must be removed from the set of candidate arms by the end of that iteration.

Hence, the number of data point computations $M_i$ required for any arm $i \neq i^*$ is at most

$$M_i \leq \min\left[\frac{16c_0^2}{\Delta_i^2}\log\frac{n}{\delta\Delta_i} + 1, 2d\right]$$

where we used the fact that the maximum number of computations for any arm is $2d$ when sampling with replacement. Note that bound this holds simultaneously for all arms $i$ with probability at least $1 - \delta$. We conclude that the total number of arm pulls $M$ satisfies

$$M \leq \sum_{i\in[n]}\min\left[\frac{16c_0^2}{\Delta_i^2}\log\frac{n}{\delta\Delta_i} + 1, 2d\right]$$

with probability at least $1 - \delta$.

As argued before, since each arm pull involves an $O(1)$ computation, $M$ also corresponds to the total number of operations up to a constant factor. $\square$

**Proof of Theorem 2**

*Proof.* Since all the $X_{iJ}$'s are unbiased, optimizing Problem (3) is equivalent to minimizing the combined second moment

$$\sum_{i \in [n]} \mathbb{E}_{J \sim P_{\mathbf{w}}}[X_{iJ}^2] = \sum_{i \in [n]} \sum_{j \in [d]} \frac{1}{d^2 w_j} q_j^2 v_{ij}^2 \qquad (5)$$

$$= \sum_{j \in [d]} \left( \frac{1}{d^2 w_j} q_j^2 \sum_{i \in [n]} v_{ij}^2 \right). \qquad (6)$$

The Lagrangian is given by

$$\mathcal{L}(\mathbf{w}, \nu) = \sum_{j \in [d]} \left( \frac{1}{d^2 w_j} q_j^2 \sum_{i \in [n]} v_{ij}^2 \right) + \nu \left( 1 - \sum_{j \in [d]} w_j \right). \qquad (7)$$

Furthermore, the derivatives are

$$\frac{\partial \mathcal{L}(\mathbf{w}, \nu)}{\partial w_j} = -\frac{q_j^2 \sum_{i \in [n]} v_{ij}^2}{d^2 w_j^2} - \nu \qquad (8)$$

$$\frac{\partial \mathcal{L}(\mathbf{w}, \nu)}{\partial \mu} = 1 - \sum_{j \in [d]} w_j. \qquad (9)$$

By the Karush-Kuhn-Tucker (KKT) conditions, setting the derivatives to 0 gives

$$w_j^* = \frac{\sqrt{q_j^2 \sum_{i \in [n]} v_{ij}^2}}{\sum_{j \in [d]} \sqrt{q_j^2 \sum_{i \in [n]} v_{ij}^2}} \quad \text{for } j = 1, \ldots, d. \qquad (10)$$

$\square$

## 9 Description of Datasets

Here, we provide a more detailed description of the datasets used in our experiments.

### Synthetic Datasets

In the NORMAL_CUSTOM dataset, a parameter $\theta_i$ is drawn for each atom from a standard normal distribution, then each coordinate for that atom is drawn from $\mathcal{N}(\theta_i, 1)$. The signals are generated similarly.

In the CORRELATED_NORMAL_CUSTOM dataset, a parameter $\theta$ is for the signal $\mathbf{q}$ from a standard normal distribution, then each coordinate for that signal is drawn from $\mathcal{N}(\theta, 1)$. Atom $\mathbf{v}_i$ is generated by first sampling a random weight $w_i \sim \mathcal{N}(0, 1)$; then atom $\mathbf{v}_i$ is set to $w_i \mathbf{q}$ plus Gaussian noise.

Note that for the synthetic datasets, we can vary $n$ and $d$. The values of $n$ and $d$ chosen for each experiment are described in Subsection 9.

### Real-world datasets

**Netflix Dataset:** We use a subset of the data from the Netflix Prize dataset (Bennett, Lanning, and Netflix 2007) that contains the ratings of 6,000 movies by 400,000 customers. We impute missing ratings by approximating the data matrix

via a low-rank approximation. Specifically, we approximate the data matrix via a 100-factor SVD decomposition. The movie vectors are used as the query vectors and atoms and $d$ corresponds to the number of subsampled users.

**Movie Lens Dataset:** We use Movie Lens-1M dataset (Harper and Konstan 2015), which consists of 1 million ratings of 4,000 movies by 6,000 users. As for the Netflix dataset, we impute missing ratings by obtaining a low-rank approximation to the data matrix. Specifically, we perform apply a Non-negative Matrix Factorization (NMF) with 15 factors to the dataset to impute missing values. The movie vectors are used as the query vectors and atoms, with $d$ corresponding to the number of subsampled users.

We note that for all datasets, the coordinate-wise inner products are sub-Gaussian random variables. In particular, this means the assumptions of Theorem 1 are satisfied and we can construct confidence intervals that scale as $O\left( \sqrt{\frac{\log 1/\delta'}{d'}} \right)$. We describe the setting for the sub-Gaussianity parameters in Section 9.

### Experimental Settings

**Scaling Experiments:** In all scaling experiments, $\delta$ and $\epsilon$ were both set to $0.001$ for BanditMIPS and BanditMIPS-$\alpha$. $\epsilon$ is the hyperparameter in bandit algorithms that controls how far the returned arm is from the true optimal arm, allowing for an $\epsilon$-suboptimal choice. For the NORMAL_CUSTOM and CORRELATED_NORMAL_CUSTOM datasets, the sub-Gaussianity parameter was set to $1$. For the Netflix and Movie Lens datasets, the sub-Gaussianity parameter was set to $25$. For the CryptoPairs, SIFT-1M, and SimpleSong datasets described in Appendix 10, the sub-Gaussianity parameters were set to $2.5e9$, $6.25e5$, and $25$, respectively. The number of atoms was set to $100$ and all other atoms used default values of hyperparameters for their sub-Gaussianity parameters.

**Tradeoff Experiments:** For the tradeoff experiments, the number of dimensions was fixed to $d = 10,000$. The various values of speedups were obtained by varying the hyperparameters of each algorithm. For NAPG-MIPS and HNSW-MIPS, for example, $M$ was varied from 4 to 32, $ef\_constructions$ was varied from 2 to 500, and $ef\_searches$ was varied from 2 to 500. For Greedy-MIPS, $budget$ varied from 2 to 999. For LSH-MIPS, the number of hash functions and hash values vary from 1 to 10. For H2ALSH, $\delta$ varies from $\frac{1}{2^4}$ to $\frac{1}{2}$, $c_0$ varies from 1.2 to 5, and $c$ varies from 0.9 to 2. For NEQ-MIPS, the number of codewords and codebooks vary from 1 to 100. For BanditMIPS, BanditMIPS-$\alpha$, and BoundedME, speedups were obtained by varying $\delta$ from $\frac{1}{10^{10}}$ to $0.99$ and $\epsilon$ from $\frac{1}{10^{10}}$ to 3. In our code submission, we include a one-line script to reproduce all of our results and plots.

All experiments were run on a 2019 Macbook Pro with a 2.4 GHz 8-Core Intel Core i9 CPU, 64 GB 2667 MHz DDR4 RAM, and an Intel UHD Graphics 630 1536 MB graphics card. Our results, however, should not be sensitive to hardware, as we used hardware-independent performance

Table 5: Frequencies for various musical notes.

| Note | Frequency (Hz) |
|------|----------------|
| C4   | 256            |
| E4   | 330            |
| G4   | 392            |
| C5   | 512            |
| E5   | 660            |
| G5   | 784            |

metrics (the number of coordinate-wise multiplications) for our results.

## 10  Application to Matching Pursuit on the `SimpleSong` Dataset

We construct a simple synthetic dataset, titled the `SimpleSong` Dataset where the query and atoms are audio signals sampled at 44,100 Hz and each coordinate value represents the signal's amplitude at a given point in time. Common musical notes are represented as periodic sine waves with the frequencies given in Table 5.

The query in this dataset is a simple song. The song is structured in 1 minute intervals, where the first interval – called an A interval – consists of a C4-E4-G4 chord and the second interval – called a B interval – consists of a G4-C5-E5 chord. The song is then repeated $t$ times, bringing its total length to $2t$ minutes. The dimensionality of the the signal is $d = 2t * 44,100 = 88,200t$. The weights of the C4, E4, and G4 waves in the A intervals and the G4, C5, and E5 waves in the B intervals are in the ratio 1:2:3:3:2.5:1.5.

The atoms in this dataset are the sine waves corresponding to the notes with the frequencies show in Table 5, as well as notes of other frequencies.

The Matching Pursuit problem (MP) is a problem in which a vector $\mathbf{q}$ is approximated as a linear combination of the atoms $\mathbf{v}_1, \ldots, \mathbf{v}_n$. A common algorithm for MP involves solving MIPS to find the atom $\mathbf{v}_{i^*}$ with the highest inner product with the query, subtracting the component of the query parallel to $\mathbf{v}_{i^*}$, and re-iterating this process with the residual. Such an approach solves MIPS several times as a subroutine. Thus, an algorithm which accelerates MIPS should also then accelerate MP.

In the audio domain, we note that when the atoms $\mathbf{v}_1, \ldots, \mathbf{v}_n$ are periodic functions with predefined frequencies, MP becomes a form of Fourier analysis in which the atoms are the Fourier components and their inner products with the query correspond to Fourier coefficients. For more detailed background on Fourier theory, we refer the reader to (Brigham 1988).

For convenience, we restrict $t$ to be an integer in our experiments so a whole number of AB intervals are completed. We ran BanditMIPS with $\delta = \frac{1}{10,000}$ and $\sigma^2 = 6.25$ over 3 random seeds for various values of $t$. BanditMIPS is correctly able to recover the notes played in the song in order of decreasing strength: G4, C5, E4, E5, and C4 in each experiment.

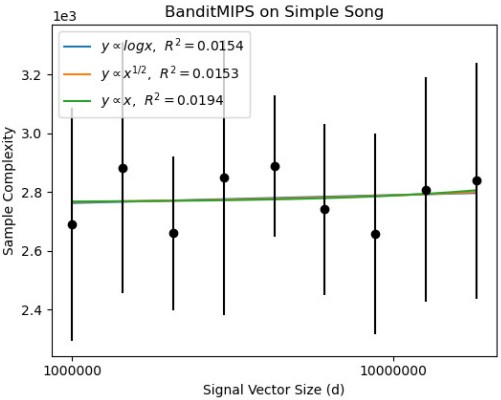

Figure 8: Sample complexity of MP when using BanditMIPS as a subroutine for MIPS on the `SimpleSong` dataset. The complexity of solving the problem does not scale with the length of the song, $d$. Uncertainties and means were obtained from 3 random seeds. BanditMIPS returns the correct solution to MIPS in each trial.

Furthermore, BanditMIPS is able to calculate their Fourier coefficients correctly. Crucially, the complexity of BanditMIPS to identify these components does not scale with $d$, the length of the song. Figure 8 demonstrates the total sample complexity of BanditMIPS to identify the first five Fourier components (five iterations of MIPS) of the song as the song length increases.

Our approach may suggest an application to Fourier transforms, which aim to represent signals in terms of constituent signals with predetermined set of frequencies. We acknowledge, however, that Fourier analysis is a well-developed field and that further research is necessary to compare such a method to state-of-the-art Fourier transform methods, which may already be heavily optimized or sampling-based.

## 11  BanditMIPS on a Highly Symmetric Dataset

In this section, we discuss a dataset on which the assumptions in Section 4 fail, namely when $\Delta$ scales with $d$. In this setting, BanditMIPS does not scale as $O(1)$ and instead scales linearly with $d$, as is expected.

We call this dataset the `SymmetricNormal` dataset. In this dataset, the signal has each coordinate drawn from $\mathcal{N}(0, 1)$ and each atom's coordinate is drawn i.i.d. from $\mathcal{N}(0, 1)$. Note that all atoms are therefore symmetric *a priori*.

We now consider the quantity $\Delta_{i,j}(d) \coloneqq \mu_1(d) - \mu_2(d)$, i.e., the gap between the first and second arm, where our notation emphasizes we are studying each quantity as $d$ increases. Note that $\Delta_{i,j}(d) = \frac{\mathbf{v}_1^T q - \mathbf{v}_2^T q}{d}$. By the Central Limit Theorem, the sequence of random variables $\sqrt{d}\Delta_{i,j}(d)$ converges in distribution to $\mathcal{N}(0, \sigma_{i,j}^2)$ for some constant $\sigma_{i,j}^2$. Crucially, this implies that $\Delta_{i,j}(d)$ is on the order of $\frac{1}{\sqrt{d}}$.

The complexity results from Theorem 1 then predicts that BanditMIPS scales linearly with $d$. Indeed, this is what we

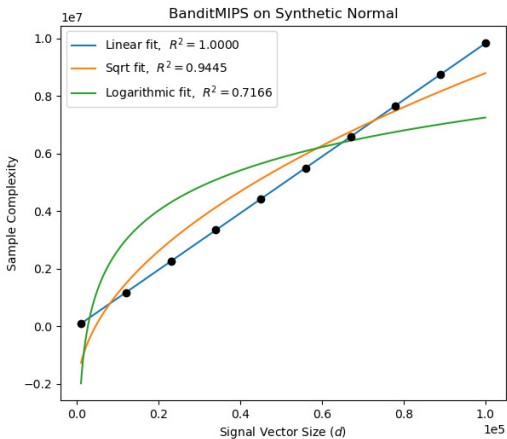

Figure 9: Sample complexity of BanditMIPS on the `SymmetricNormal` dataset. The sample complexity of BanditMIPS is linear with $d$, as is expected. Uncertainties and means were obtained from 10 random seeds.

observe in Figure 9.

In practice, this case can be dealt with by allowing for an $\epsilon$-suboptimal atom vector to be returned. In this case, BanditMIPS will no longer depend on the $\Delta_i$'s for large $d$, and instead on the relative error hyperparameter $\epsilon$. This is depicted in figure 10.

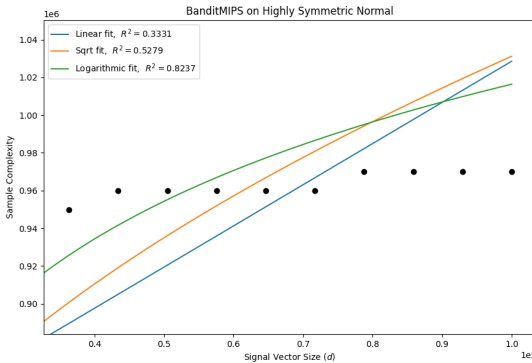

Figure 10: Sample Complexity of BanditMIPS on the Highly Symmetric Normal Dataset as a Function of $d$, allowing for $\epsilon$-suboptimal atoms to be identified, with $\epsilon = 0.1$. The sample complexity of BanditMIPS scales as $O(1)$ with respect to $d$, even when all atoms have an equal inner product with the query vector.