# OpenReview forum: "Faster Maximum Inner Product Search in High Dimensions"
_ICLR.cc/2024/Conference — ICLR 2024 Conference Withdrawn Submission_

### Official Review · Reviewer_tsE3 · 2023-10-23

**Soundness:** 2 fair
**Presentation:** 2 fair
**Contribution:** 3 good
**Rating:** 3
**Confidence:** 3

**Summary:**

The paper studies the maximum inner product search (MIPS) problem in high dimensions. The authors present a multi-armed bandit-based approach, called BanditMIPS, whose idea is to adaptively subsample coordinates for more promising atoms. Empirically, BanditMIPS can achieve 20 times faster than the prior algorithm.

**Strengths:**

- The MIPS problem is important and relevant. A fast algorithm for this problem is useful.
- The introduction of the multi-armed bandit idea is natural.
- The empirical results look convincing.

**Weaknesses:**

- The format of the paper is incorrect.
- The theoretical result is as simple as a standard corollary of UCB bounds.
- The paper only studies the runtime for a single query vector, which is not sufficient in practice since we usually face multiple queries.

**Questions:**

- The paper discusses a single query vector and hence, mentions that BanditMIPS requires no preprocessing of the data. However, we note that there is usually a long sequence of queries in practice, which requires some preprocessing, e.g., HNSW. A discussion of the two settings maybe helpful.

---

### Official Review · Reviewer_CJG9 · 2023-10-28

**Soundness:** 1 poor
**Presentation:** 1 poor
**Contribution:** 1 poor
**Rating:** 1
**Confidence:** 4

**Summary:**

The paper does not follow the ICLR formatting guidelines (the paper is two columns etc) and does not have any references in the main submission. Even ignoring this, I'm not sure what the main theorem is even saying. Roughly it states that 'if some parameter is small enough', then we only need to read few coordinates for maximum inner product search. But the 'parameter' in question seems to be defined circularly as 'how many coordinates are read'.

**Strengths:**

See summary above.

**Weaknesses:**

See summary above.

**Questions:**

See summary above.

---

### Official Review · Reviewer_FGUA · 2023-11-01

**Soundness:** 2 fair
**Presentation:** 2 fair
**Contribution:** 2 fair
**Rating:** 3
**Confidence:** 3

**Summary:**

This paper studies the problem of Maximum Inner Product Search problem (MIPS): Given a dataset of n vectors in a d-dimensional space, and a new vector, the goal is to find the vector within the dataset that yields the highest inner product with it. This paper introduces a randomized algorithm inspired by bandit methods to enhance the time complexity from $O_n(\sqrt{d})$ to $O_n(1)$, and showcases its effectiveness on four synthetic and real-world datasets.

**Strengths:**

- the problem studied in this paper is interesting
- the paper is well-organized, and easy to follow.
- there are extensive evaluations of their methods over various datasets

**Weaknesses:**

- The paper assumes that $X_i$ follows subgaussian distribution, this is a strong assumption.
- The presentation of this paper needs major improvement. Some of the figures are super large(Figure 3,5,6), while some of the figures are extra small to read(Figure 4).

Also, the current format of this paper doesn't align with the ICLR 2024 format file, and the references are not in the submitted 9 pages.

Minor Comments:
- Duplicate reference (in the appendix version):
Morozov, S.; and Babenko, A. 2018a. Non-metric Similarity Graphs for Maximum Inner Product Search.
Morozov, S.; and Babenko, A. 2018b. Non-Metric Similarity Graphs for Maximum Inner Product Search. In Advances in Neural Information Processing Systems, volume 31. Curran Associates, Inc.
- "Accuracy:1.0" in figure 5 and figure 6 is not very clear to read and isn't been explained in the caption.
- subplots in Figure 4 lacks effort, note that if we print this paper out, these subgraphs won't be big enough to read.

**Questions:**

see weakness on the subgaussian distribution.

---

### Official Review · Reviewer_hfLb · 2023-11-03

**Soundness:** 3 good
**Presentation:** 3 good
**Contribution:** 1 poor
**Rating:** 1
**Confidence:** 5

**Summary:**

This work considers the problem of maximum inner product search (MIPS), where given a static database v_1...v_n of n points, each in $\mathbb{R}^d$, and a query point $q$, asks to find the point $v^*$ in the database with maximum inner product $v^*= argmax_{i\in [n]} <q,v_i>$ with the query point. This is a practically important problem, with the trivial solution requiring $O(nd)$ computations. The key question here is can we speed up this search, i.e. identify the point $v^*$ in far fewer than $O(nd)$ operations. In this paper, the authors derive a scheme by reducing this problem to the best arm identification problem in stochastic bandits, by essentially treating each point in the static database as an arm, with mean reward of arm $i$ corresponding to the database vector $v_i$ effectively being $\sum_{j=1}^d q_j\cdot v_{ij}/d$. The "stochastic" component comes from the fact the we can obtain a noisy estimate of this quantity in $O(1)$ time by sampling a dimension $j\in [d]$ uniformly at random (with replacement) and computing the product $q_jv_{ij}$. This problem then exactly becomes the best arm identification problem, with the total computational complexity of MIPS being the sample complexity of the corresponding best arm identification problem. Following standard results in the BAI literature, the authors prove that MIPS can be solved with high probability in $O(\sum_{I=2}^n \frac{1}{\Delta_i^2}\log (n/\delta))$ computations, which is dimension invariant.

**Strengths:**

None.

**Weaknesses:**

I apologize for the harshness in this review, but I strongly believe it is needed with this submission. There is absolutely nothing technically interesting going on in this paper. The idea of using best arm identification to solve MIPS is very obvious and has already been explored before in the result of Liu et.al. AAAI 19, with the only difference being sampling with vs without replacement. The statement that they achieve dimension independent bounds is a huge overclaim -> the dependence on $1/\Delta_i^2$ directly scales with the dimension, and you explicitly need well-separation assumptions in order to get around this (or settle for an $\epsilon$-approximate solution which then directly bounds this quantity by $1/\epsilon^2$). None of these ideas are at all novel, due to which for this result, I find it extremely difficult to recommend an accept at any venue, let alone one as competitive as ICLR.

**Questions:**

What is novel in this result?